# Simulation and Experimental Research on a Beam Homogenization System of a Semiconductor Laser

**DOI:** 10.3390/s22103725

**Published:** 2022-05-13

**Authors:** Haijing Zheng, Huayan Sun, Huaili Zhang, Yingchun Li, Huichao Guo, Laixian Zhang, Rong Li, Qiang Yin

**Affiliations:** 1Department of Electronic and Optical Engineering, Space Engineering University, Beijing 101407, China; shy221528@vip.sina.com (H.S.); zhj_sea@hotmail.com (H.Z.); 13811420115@163.com (Y.L.); rongli117@yeah.net (H.G.); zhanglaixian@126.com (L.Z.); bitlirong@163.com (R.L.); 2Xi’an Satellite Control Center, Xi’an 710043, China; yinqiang106335@163.com

**Keywords:** semiconductor laser, laser active imaging detection, homogenizing pipe, laser illumination system

## Abstract

Aiming at the application of laser active imaging detection technology, this paper studied the beam homogenization system of a semiconductor laser based on a homogenizing pipe. Firstly, the principle of the homogenizing pipe was introduced. Secondly, the homogenization effect, which was influenced by several geometric parameters (aperture size, length, and taper) of the homogenizing pipe using the optical design software, was simulated for the fiber-coupled semiconductor laser. Finally, according to the simulated results, a laser illumination system composed of a fiber-coupled semiconductor laser, a homogenizing pipe, and an aspheric lens was designed, which can obtain a rectangular uniform light spot in a long distance. The effectiveness of the illumination system was verified by simulation and experiment, respectively. Simulation results suggested that the uniformity of the spot at a distance of 20 m was 85.6%, while divergence angle was 10 mrad. The uniformity of the spot at a distance of 120 m was 91.5%, while divergence angle was 10 mrad. Experimental results showed that the uniformity of the spot at a distance of 20 m was 87.7%, while divergence angle was 13 mrad. The uniformity of the spot at a distance of 120 m was 93.3%, while divergence angle was 15 mrad. The laser illumination system designed in this paper was simple and easy to assemble, and has strong practicability. The results in this paper have certain reference value and guiding significance for the homogenization design of semiconductor lasers.

## 1. Introduction

Semiconductor lasers are small in size, long in life, and high in photoelectric conversion efficiency. They have been widely used in communication, optical storage, LiDAR, aerospace, material cutting and welding, lighting, and other fields. However, the beam quality of semiconductor lasers is poor, in which energy distribution of the far-field spot is elliptical Gaussian. Moreover, there is intrinsic astigmatism. In applications such as laser welding, cladding, and surface heat treatment, uneven energy distribution can easily lead to excessive local temperature of the material, and affect the performance of the material. In the field of photocatalysis, ultraviolet laser light sources have been paid attention to because of a series of advantages. Uniform catalytic effect can be obtained by using the UV semiconductor laser light source with uniform energy distribution. In the wide application of semiconductor laser therapy instruments, the light spot with uniform output energy can increase the effective treatment area of laser irradiation. Thus, the treatment effect is more significant. In the field of illumination, the laser beam needs to be evenly projected onto the target surface to obtain better imaging quality. In many applications of semiconductor lasers, the requirements of energy homogenization have been put forward for their beams [1,2].

Laser active imaging detection technology is an active imaging technology which uses a laser beam as the light source with specific parameters in a specific band, which irradiates the target. Then, the reflected laser beam from the target is detected. Finally, the target’s image is obtained [3]. In the case of severe weather conditions and strong medium scattering, laser active imaging detection technology can effectively avoid the interference of backscattered beams. It can image under low illumination and complex background conditions while obtaining not only the range image but also the intensity image of the target. This technology has the irreplaceable advantage of passive imaging systems. The illumination system is an important part of the laser active imaging detection system. Due to the beam transmission characteristics of semiconductor lasers, it is necessary to design suitable illumination systems for different types of semiconductor lasers. In order to meet the application requirements of the laser active imaging detection system, the output beam of the semiconductor laser must be homogenized.

Common beam homogenization methods include: aspheric lens method, diffractive optical element method, fiber waveguide method, and microlens array method, etc. [4,5]

The homogenization principle of the aspheric lens method is to use the phase modulation effect of the aspheric lens on the light, which was proposed by Frieden in 1965 [6]. Through the law of conservation of light energy and the law of equal optical path, various parameters of the aspheric lens can be calculated and obtained. The advantage of the aspheric lens method is that the structure is relatively simple and the loss of light energy is small. For single-mode laser beams, better homogenization results can be obtained by this method. Since a set of aspherical lenses can only be homogenized for a laser beam with a fixed light energy distribution, this method is not universal, and it is difficult to manufacture [7].

The diffractive optical element is etched on the lens surface by mask processing technology to obtain a step structure. The diffractive optical element is a pure phase optical element with very high diffraction efficiency. The homogenization effect and energy utilization rate obtained by this method are at a high level. However, the design and the manufacturing difficulty of this method are relatively high [8,9,10,11].

The microlens array method achieves the homogenization effect of the semiconductor laser beam by using one or two microlens arrays and a Fourier lens. It can also achieve the function of shaping. Compared with other methods, the uniformity of the beam after homogenization by this method is higher. However, at the same time, the system structure is relatively complex while the light energy loss is large after multiple lenses [12,13,14].

The optical fiber waveguide method uses the principle of total reflection of light to reflect the light beam coupled into the fiber multiple times. This method obtains a uniformly light spot on the exit end face of the optical fiber. The optical fiber waveguide method has a relatively simple structure which is easy to implement. The homogenizing pipe is one of the most typical fiber waveguides [15,16,17,18,19,20,21]. M. Traub simulated the intensity distributions changing with the number of reflections inside the homogenizing pipe [15]. However, the aperture of light source and the taper of the homogenizing pipe were not considered. M. Laurenzis presented two different illumination devices based on the homogenizing pipe for a solid-state laser source and an array of semiconductor laser diodes [22]. The work lacked analysis for the influence of different parameters of the homogenizing pipe. Y. Song presented the work of two-dimensional beam shaping and homogenization of a high power laser diode stack by a rectangular waveguide [23]. However, the special-shaped homogenizing pipe was not discussed.

Most of the literature focuses on regular homogenizing pipes, such as the rectangular homogenizing pipe. Additionally, the influence of assembly caused by various parameter changes of the homogenizing pipe was not studied. In this paper, aiming at the application of laser active imaging detection technology, the research on beam homogenization of a semiconductor laser is carried out based on the homogenizing pipe. Firstly, the principle of total reflection of the homogenizing pipe is deeply explored, and the influence of the cross-sectional shape of the homogenizing pipe on the homogenization effect is discussed. Secondly, for the fiber-coupled semiconductor laser, the optical design software is used to simulate and analyze the influence of geometric parameters such as aperture size, length, and taper of the homogenizing pipe on the homogenization effect. According to the analysis results, a laser beam homogenizing system is designed, which is the illumination system. Finally, the effectiveness and practicability of the system are verified by simulation and experiment, respectively.

## 2. Methods

### 2.1. Spot Quality Evaluation

To study the beam homogenization method, it is necessary to introduce evaluation parameters to measure the uniformity of the spot. At present, the evaluation methods used to measure beam uniformity mainly include Root Mean Square-*M*_RMS_, Mean Absolute Error-*γ* [24], Evaluation Based on Stability, etc. Since the Mean Absolute Error-γ can make an overall description of the uniformity of light intensity, which is simple and convenient to use, this paper chose the Mean Absolute Error-*γ* as the evaluation method for the uniformity of the light spot.

Mean Absolute Error-*γ* is defined as
(1)$$\gamma \;=\;\left(1\;-\;\frac{\sum \left|I_{i}-\overset{-}{I}\right|}{N\;\times \;\overset{-}{I}}\right)\;\times \;100\%$$
in which, *I_i_* is the intensity of each sampling point of the light spot, $\overset{-}{I}$ is the average value of intensity for all sampling points, and *N* is the number of sampling points. The closer the *γ* is to 1, the better the beam homogenization effect is.

### 2.2. Principle of Homogenizing Pipe

The homogenizing pipe, also known as the kaleidoscope, was invented by the Scottish physicist David Brewster in 1814. Its most basic property is to satisfy the law of refraction and reflection of the light beam. The symmetric image of reflections circulating between the mirrors constitutes the most magical aspect of the kaleidoscope [25]. Figure 1 shows the image formed by a typical kaleidoscope.

The homogenizing pipe is the most common form of the optical fiber waveguide method. Light propagates inside the homogenizing pipe based on the principle of total reflection of light. The inside wall of the homogenizing pipe is coated with a high-reflection film. When the incident light is introduced into the homogenizing pipe, it undergoes multiple total reflections inside the homogenizing tube. Finally, light beams exit at the other end of the homogenizing pipe to obtain a uniform light spot, of which the shape is consistent with the exit cross-section of the homogenizing pipe. Figure 2 shows the light propagation route in the homogenizing pipe.

The cross-section of the homogenizing pipe can be designed in various shapes. In general, triangular, rectangular, hexagonal, and other end faces can achieve better uniformity. Meanwhile, a round exit face has poor uniformity. In many optical systems, a taper homogenizing pipe is also used, which can change the divergence angle of the output light to meet the requirements of the numerical aperture of the subsequent system. For a homogenizing pipe with a specific surface, and a beam with a specific spatial angle distribution, the longer the homogenizing pipe, the more reflection times, consequently, the more uniform the light intensity distribution on the exit end face. However, for a regular n-shaped homogenizing pipe with more end faces than hexagons, since the imaging distributions overlap in space, the light intensity distribution on the exit end face is not necessarily more uniform than that of a square homogenizing pipe. 

### 2.3. Simulation

The design of the illumination system in the laser active imaging detection system should comprehensively consider several factors, such as the characteristics of the laser beam, the structural size of the illumination system, the cost and difficulty of the optical lens manufacturing, and the error of the system assembly and debugging.

#### 2.3.1. Light Source

The light source used in this work was a fiber-coupled semiconductor laser. Fiber-coupled semiconductor lasers refer to semiconductor laser modules that shape the laser beam and focus it into the fiber for transmission after satisfying the fiber coupling conditions. Fiber-coupled semiconductor lasers can realize the flexible transmission of the light beam, as well as the high power and high brightness output laser beam, which has very high value for the application of the semiconductor laser. The applications of fiber-coupled semiconductor lasers mainly include pumping, industrial processing, medical cosmetology, and military. These have developed rapidly in all walks of life and have gradually become a vital device in the field of lasers.

The fiber bundle of the fiber-coupled semiconductor laser used in this work consists of 37 fibers. Its exit end face is a regular hexagonal array, as shown in Figure 3. The diameter of the circumscribed circle is 2 mm, while the light beam divergence angle is 22°.

The optical design software was used to simulate the light spots formed by this light source at different distances, as shown in Figure 4. Figure 4a is the light spot when the distance is 0.05 mm, and Figure 4b is the light spot when the distance is 5 mm. It can be seen that the far-field spot energy distribution of the light source is Gaussian type, in which uniformity was very poor. Therefore, uniform light treatment was required.

#### 2.3.2. Homogenizing Pipe

When the light beam was transmitted inside the homogenizing pipe, it should satisfy the condition of total reflection. Figure 5 shows the schematic diagram of the total reflection in the homogenizing pipe, in which *θ* means beam divergence angle, and *i* means reflection angle.

If the material of the homogenizing pipe is Silica and the internal refractive index *n* = 1.458464, the critical angle of total reflection is
(2)$$I_{C}\;=\;arcsin\left(\frac{1}{n}\right),$$

The condition for total reflection is $90{}^{\circ}\;\leq \;i\;\leq \;i_{c}$. Due to θ = 90° − i, then *θ* satisfies
(3)0° ≤ θ ≤ 46.72°,

Therefore, as long as Formula (3) is satisfied, the light beam can be totally reflected inside the homogenizing pipe.

Since the divergence angle of the output beam of the fiber-coupled semiconductor laser was 22°, which satisfied Formula (3), the exit end of the fiber was directly coupled to the entrance end of the homogenizing pipe without adding a coupling lens. Figure 6 shows a schematic diagram of the direct coupling between the optical fiber and the homogenizing pipe, in which the aperture of the homogenizing pipe is square. Figure 7 shows the light spot at the exit end of the homogenizing pipe.

Comparing the light spot before and after the homogenizing pipe, it can be seen that the light spot is homogenized and shaped from the hexagonal array shown in Figure 4a into a uniformly distributed rectangular spot as shown in Figure 7. The simulation results suggested that the homogenizing pipe has the effect of shaping and homogenizing at the same time.

When designing the homogenizing pipe, three parameters were mainly considered: the aperture size, the length, and the taper.

### 2.4. Experiment

Figure 8 is the schematic diagram of the experimental system. The experimental system can be divided into an illumination system and an imaging detection system. The illumination system consisted of a fiber-coupled semiconductor laser, a homogenizing pipe, and an aspheric lens.

#### 2.4.1. Illumination System

The power of the fiber-coupled laser was 500 W, in which the wavelength was 806 nm. The diameter of the outlet end face was 2 mm, and the initial divergence angle was 22°. Figure 9 shows the fiber-coupled semiconductor laser outlet end.

The homogenizing pipe was customized according to the parameters determined in Section 3. In order to analyze the attenuation of the laser beam by the homogenizing pipe, two homogenizing pipes with different lengths were customized in this experiment. Table 1 shows the parameters of the two homogenizing pipes.

Solebo’s AL5040-B aspherical lens was selected as the beam collimation device. This aspheric lens has an effective focal length of 40 mm, a numerical aperture of 0.55, an outer diameter of 50 mm, a working distance of 31.3 mm, and a central thickness of 10.0 mm. Its material was S-LAH64. Figure 10 shows the image of the AL5040-B aspheric lens.

Figure 11 shows the experimental setup of the illumination system. The fiber-coupled semiconductor laser emitted a laser beam with a Gaussian distribution, which passed through a homogenizing pipe and an aspherical lens in turn. Then a light spot with uniform energy distribution was obtained in the far field.

#### 2.4.2. Imaging Detection System

The illumination system emitted a laser beam and irradiated it on the target. The imaging detection system received the returned light beam. Figure 12 shows the image of the detection system. This detection system consisted of an 85~200 mm zoom lens, CCD, power supply, and a video capture card.

## 3. Results and Discussion

### 3.1. Simulation Results

#### 3.1.1. Influence of Aperture Size

In order to analyze the influence of the aperture size of the homogenizing pipe on the homogenization effect, the parameter Super Source Ratio (SSR) was defined, which expresses the proportional relationship between the aperture size of homogenizing pipe and the aperture size of the light source. The *SSR* is defined as
(4)$$SSR\;=\;\frac{D_{T}}{D_{S}}\;-\;1,$$
in which, *D_T_* means the size of homogenizing pipe aperture and *D_S_* means the size of the light source aperture.

It is assumed that the inlet and outlet end of the homogenizing pipe are rectangles with the same size, while the length of the homogenizing pipe is 20 mm, and the diameter of the light source is 2 mm. The change of spot uniformity at the outlet end of the pipe is simulated when the SSR increases from 0.25 to 5 (the size of the pipe’s aperture from 0.5 mm × 0.5 mm to 10 mm × 10 mm). Figure 13 shows the light spot at the exit end of the homogenizing pipe under different SSRs, in which the image of the light spot is zoomed.

In order to quantitatively evaluate the homogenization effect of the homogenizing pipe on the light beam, the Average Absolute Error-γ was used as the evaluation index of the uniformity of the light spot. Hence, the uniformity of the light spot was calculated by Formula (1). Figure 14 shows the variation curve of the uniformity at the exit end of the homogenizing pipe with the SSR.

The above simulation results demonstrated that when the aperture size of the pipe gradually increased, the uniformity of the light spot gradually increased. When the SSR reached 1, the uniformity reached a peak value of 87%. At this time, the homogenization effect of the pipe was the best. When the SSR varied from 0.8 to 3, the uniformity was above 80%.

#### 3.1.2. Influence of Length

The size of the entrance aperture and the exit aperture of the homogenizing pipe are set in a rectangle of 2 mm × 2 mm, while the material of the homogenizing pipe is Silica. The uniformity of the light spot on the exit end face of the pipe was calculated at different lengths. Figure 15 shows the variation curve of uniformity with the length of the homogenizing pipe.

It can be seen from Figure 15 that with the increase of the length of the homogenizing pipe, the uniformity of the light spot was becoming better and better due to the increase in the times of reflections of the light beam in the pipe. That is to say, the homogenizing ability of the pipe was gradually improved. Due to the influence of the uniformity of material filling and the structure of the pipe, when the length reached 20 mm, the uniformity remained stable. When continuing to increase the length, there was no significant effect on the uniformity.

It is worth mentioning that the length of 20 mm was calculated under the parameters determined by the system in this paper. The optimal lengths of other different system parameters need to be calculated separately.

#### 3.1.3. Influence of Taper

The entrance aperture of the homogenizing pipe was set to a rectangle with a diameter of 2 mm × 2 mm, while the length was set to 20 mm. The uniformity of the light spot on the exit end face of the homogenizing pipe was calculated with different tapers. Figure 16 is a schematic diagram of a special-shaped homogenizing pipe. The exit aperture of the homogenizing pipe was inconsistent with the entrance aperture, where α is the angle between the long side of the pipe and the center line. α was used to represent the taper of the homogenizing pipe. Figure 17 shows the variation curve of uniformity with the angle α.

A negative value of the angle α indicated that the inlet aperture size of the pipe was larger than the outlet aperture size, while a positive value of the angle α indicated that the inlet aperture size was smaller than the outlet aperture size. Analyzing the variation of uniformity with the angle α, it was found that as the angle α gradually increased, the spot uniformity gradually increased. When α = 0.75°, the uniformity reached the peak value of 88%; at the same time, the size of the outlet aperture was 0.96 mm × 0.96 mm. However, the uniformity gradually decreased when continuing to increase α. When α changed in the range of −0.3° to 1°, the uniformity was above 80%.

#### 3.1.4. Illumination System

It can be seen from the above discussion that when the SSR of the homogenizing pipe was 1, the homogenization effect was the best. The longer the length of the pipe, the better the homogenizing effect. However, when the length of the pipe reached 20 mm, the homogenization effect remained unchanged. When the taper was 0.75°, the homogenization effect was the best. 

The illumination system was designed for the aforementioned fiber-coupled semiconductor laser, with a diameter of 2 mm and a divergence angle of 22°. The inlet end of the homogenizing pipe was 2 mm × 2 mm. Since the shape of the detector in the laser active imaging detection system was a rectangle with an aspect ratio of 4:3, when the far-field light spot was also a rectangle with a 4:3 rectangle, the energy utilization rate was the largest. Hence, the outlet end face of the homogenizing pipe was designed to be 1.6 mm × 1.2 mm, while the length was 20 mm. Figure 18 shows the shaded model layout of the homogenizing pipe.

The uniformity beam emitted by the homogenizing pipe had a large divergence angle, so far-field illumination cannot be achieved. Therefore, a collimating element must be added after the homogenizing pipe. The collimation methods mainly included traditional macro lens method, cylindrical lens method, self-focusing lens method, aspheric lens method, liquid crystal spatial light modulator method, etc. The collimation effect of the aspheric lens was good, while the adjustment was flexible and convenient. However, the design and manufacturing was difficult. This work used a commercially available aspherical lens, ThorLabs’ AL5040M-B. 

Figure 19 shows the simulated illumination system. Figure 19a is the 2D layout. Figure 19b is the shaded model layout. The laser beam was emitted from a fiber-coupled laser. Then, it was shaped into a uniform rectangular spot with an aspect ratio of 4:3 through a homogenizing pipe. Afterwards, it was collimated by an aspherical lens. Finally, a detector was set at a distance of 20 m, 120 m, and 500 m, respectively, to receive the laser spot.

Figure 20, Figure 21 and Figure 22 show the far-field illumination effect. Figure 20 shows the laser spot with a far-field distance of 20 m and its energy distribution, in which Figure 20a shows the laser spot at 20 m, Figure 20b is the energy distribution curve in the horizontal direction of the spot, Figure 20c is the energy distribution in the vertical direction of the light spot, and Figure 20d is the 3D diagram of the energy distribution of the light spot.

The uniformity of the light spot in Figure 20a was calculated using Formula (1) which was 85.6%. In addition, as can be seen from Figure 20b,c, the spot size at distance of 20 m was 0.16 m × 0.12 m, which means the diagonal length was 0.2 m, whereupon the half-divergence angle of the illumination system was 10 mrad.

Figure 21 shows the laser spot with a far-field distance of 120 m and its energy distribution, in which Figure 21a shows the laser spot at 120 m, Figure 21b is the energy distribution curve in the horizontal direction of the spot, Figure 21c is the energy distribution in the vertical direction of the light spot, and Figure 21d is the 3D diagram of the energy distribution of the light spot.

The uniformity of the light spot in Figure 21a was calculated using Formula (1) which was 91.5%. In addition, as can be seen from Figure 21b,c, the spot size at distance of 120 m was 0.96 m × 0.72 m, which means the diagonal length was 1.2 m, whereupon the half-divergence angle of the illumination system was 10 mrad.

Figure 22 shows the laser spot with a far-field distance of 500 m and its energy distribution, in which Figure 22a shows the laser spot at 500 m, Figure 22b is the energy distribution curve in the horizontal direction of the spot, Figure 22c is the energy distribution in the vertical direction of the light spot, and Figure 22d is the 3D diagram of the energy distribution of the light spot.

Same as above, the uniformity of the light spot in Figure 22a was calculated using Formula (1) which was 95%. In addition, as can be seen from Figure 22b,c, the spot size at distance of 500 m was 8 m × 6 m, which means the diagonal length was 10 m, whereupon the half-divergence angle of the illumination system was 10 mrad.

For Figure 20, Figure 21 and Figure 22, from the results of 20 m, 120 m, and 500 m, the uniformity was improved by increasing the distance. This was due to the action of the aspherical lens. The laser beam underwent multiple total reflections inside the homogenizing pipe. After that, it was emitted from the exit end of the homogenizing pipe. Then, the diverging beam was collimated by an aspheric lens and projected to a distance. From the view point of imaging, the role of the aspheric lens was equivalent to imaging the exit end face of the homogenizing pipe at infinity. Theoretically, there was a clear image of the exit end of the homogenizing pipe at infinity. Therefore, the farther the distance was, the clearer the image, and the higher the uniformity of the light spot.

### 3.2. Experimental Results

#### 3.2.1. Energy Loss

Due to the energy loss of the laser beam when it was transmitted in the homogenizing pipe and the aspheric lens, the energy utilization rate should also be considered when selecting the homogenizing pipe. In the experiment, an optical power meter was used to measure the light power in three cases: (1) the laser outlet end; (2) only through the homogenizing pipe; (3) only through the aspheric lens. Then, the light energy transmittance of the homogenizing pipe and the aspheric lens were calculated, respectively. Table 2 shows the transmittance of the homogenizing pipe and aspheric lens.

The transmittance of an aspheric lens is mainly affected by absorption and reflection. Since this aspherical lens was coated with an anti-reflection coating, the transmittance was relatively high at 86.4%.

The transmittance of a homogenizing pipe is mainly affected by absorption. The longer the light travels inside the homogenizing pipe, the more energy is absorbed and the lower the transmittance. By comparing the transmittance of the two types of homogenizing pipes, it can be seen that the length of the homogenizing pipe has a greater influence on the utilization rate of light energy. Under the condition that other parameters remain unchanged, the longer the length of the homogenizing pipe, the greater the loss of light energy, the lower the light transmittance. Therefore, a homogenizing pipe with a shorter length can be selected without affecting other evaluation indicators.

#### 3.2.2. Far-Field Light Spot

A far-field light spot at the distance of 20 m was received using a white screen and imaged using the detection system. Figure 23 shows the light spot image at the distance of 20 m.

It can be seen from Figure 23 that the light spot at the distance of 20 m was rectangular with uniform energy distribution. The illumination effect was good. Using Formula (1) to calculate the uniformity of the light spot, its value was 87%. Measuring the diagonal length of the rectangular spot, its value was 260 mm. Therefore, the beam divergence angle is 13 mrad. The experiment results were compared with the simulation results, as shown in Table 3.

The size and uniformity of the light spot at the distance of 120 m were measured. The light spot at the distance of 120 m was so large that an ordinary white screen may not be able to receive it completely, thus, a flat large wall was used for measurement. Figure 24 shows the measurement site and the resulting light spot.

It can be seen from Figure 24 that the light spot at 120 m was also rectangular with uniform energy distribution. The illumination effect was good. Calculating the spot uniformity, its value was 93%. Measuring the diagonal length of the rectangular spot, its value was 1.8 m. Therefore the beam divergence angle was 15 mrad. The experiment results were compared with the simulation results, as shown in Table 4.

## 4. Conclusions

The simulation results revealed that the homogenizing effect of the homogenizing pipe was greatly affected by the aperture size, length, and taper. (1) Aperture size. In fact, the homogenizing effect of the homogenizing pipe was affected by the ratio of the light source size to the pipe’s aperture size, that is, SSR. For a fixed-size light source, when the SSR was less than 1, with the increase of the aperture size, the effect of the homogenizing pipe was becoming better and better. When the SSR = 1, the effect was the best. Continuing to increase the aperture size, the effect of the pipe became worse. (2) Length. As the length of the homogenizing pipe increased, the number of reflections of the beam in the homogenizing pipe increased. The homogenizing effect of the homogenizing pipe became better and better. When the homogenizing pipe reached a certain length, affected by the uniformity of material filling and its own structure, the homogenizing effect did not become better with the increase of length, but basically remained unchanged. (3) Taper. When the taper changed from a negative value to a positive value, as the taper gradually increased, the homogenizing effect of the homogenizing pipe gradually became better. When the taper increased to 0.75°, the homogenizing effect was the best. When the taper was greater than 0.75°, the homogenizing effect decreased.

Based on the above conclusions, an illumination system was designed, which consisted of a semiconductor laser, a homogenizing pipe, and an aspheric lens. The SSR of the homogenizing pipe was 1, the length was 20 mm, and the taper was 0.75°. The simulation results showed that the divergence angle of the illumination system was 10 mrad, the far-field beam spot was rectangular, and the uniformity was above 85%.

Comparing the simulation and experimental results, it can be seen that the experiment and simulation are in good agreement. The uniformity of the light spot obtained by the experiment was also above 85%. However, there was a certain error in the divergence angle between the experimental system and the simulation system. By analysis, we know that the biggest influencing factor should be the assembly error. Due to certain errors in the assembly, and the distances between the light source and the homogenizing pipe, the homogenizing pipe and the aspheric lens were not strictly consistent with the simulation parameters, which resulted in inconsistent divergence angles.

All in all, this paper analyzed the influence of the geometric parameters of the homogenizing pipe on its homogenizing effect. The conclusions obtained in this work have certain guiding significance for the design of the homogenizing pipe, which fully meets the illumination requirements of a laser active imaging system. For other applications that require a uniform laser spot, such as medical treatment and welding, the research results in this paper also have certain reference value.

## Figures and Tables

**Figure 1 sensors-22-03725-f001:**
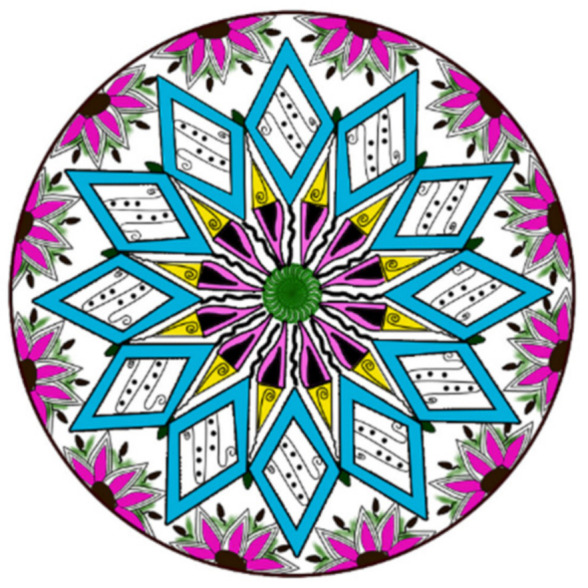
Image formed by a typical kaleidoscope.

**Figure 2 sensors-22-03725-f002:**
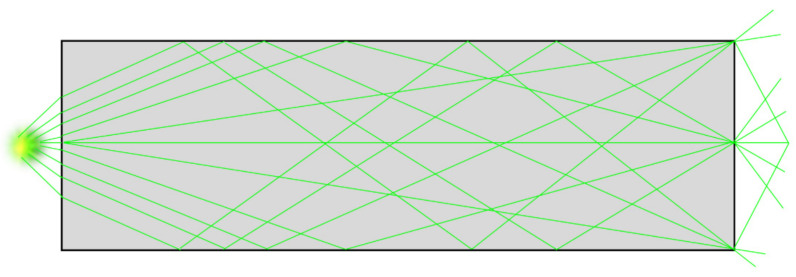
Light propagating in the homogenizing pipe.

**Figure 3 sensors-22-03725-f003:**
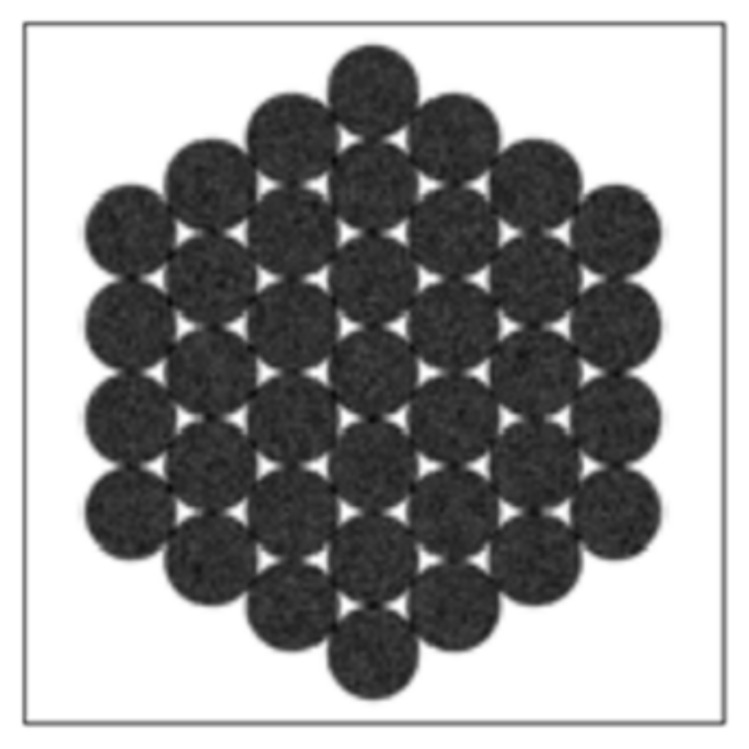
Exit end of the fiber-coupled semiconductor laser.

**Figure 4 sensors-22-03725-f004:**
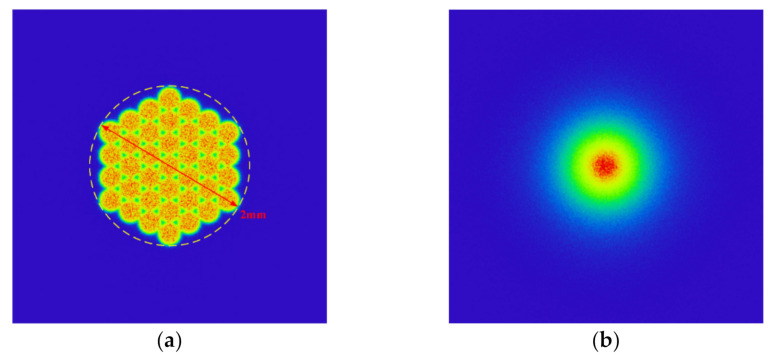
Light spots of fiber-coupled semiconductor lasers at different distances: (**a**) A distance of 0.05 mm; (**b**) A distance of 5 mm.

**Figure 5 sensors-22-03725-f005:**
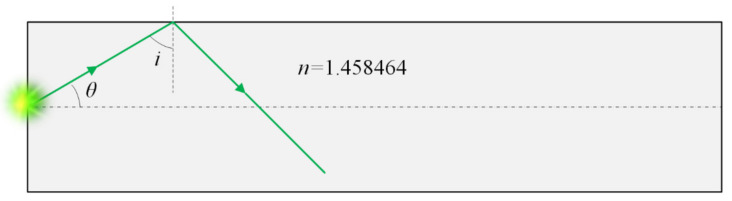
Schematic diagram of the total reflection in the homogenizing pipe.

**Figure 6 sensors-22-03725-f006:**
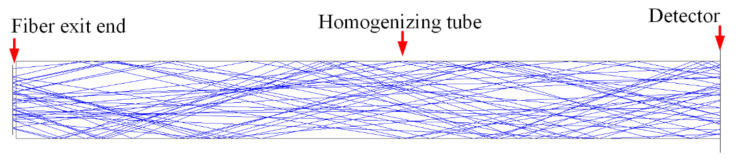
Schematic diagram of direct coupling between optical fiber and homogenizing pipe.

**Figure 7 sensors-22-03725-f007:**
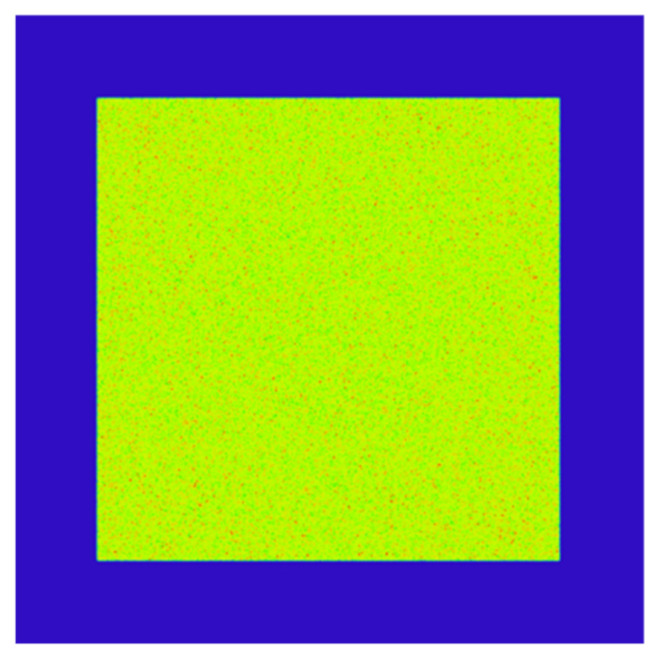
Light spot at the exit end of the homogenizing pipe.

**Figure 8 sensors-22-03725-f008:**
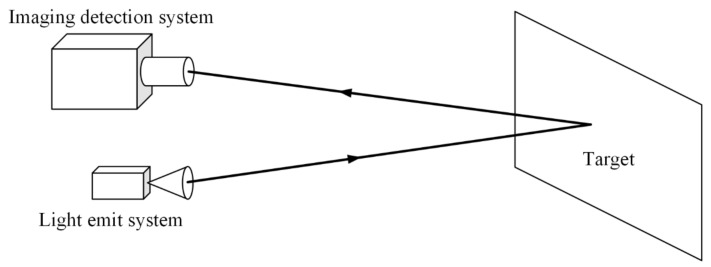
Schematic diagram of the experimental system.

**Figure 9 sensors-22-03725-f009:**
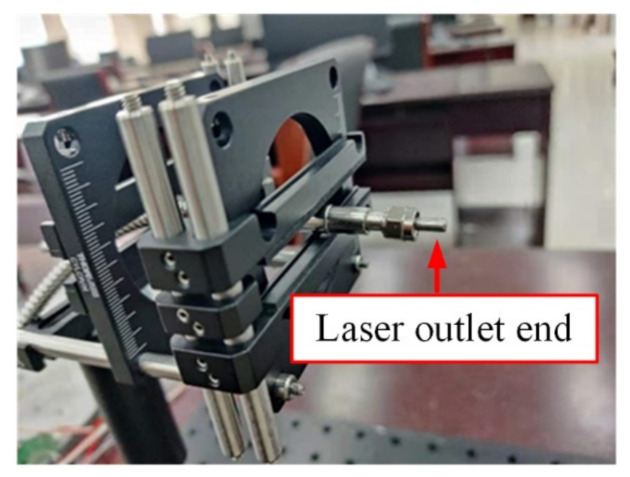
Fiber-coupled semiconductor laser outlet end.

**Figure 10 sensors-22-03725-f010:**
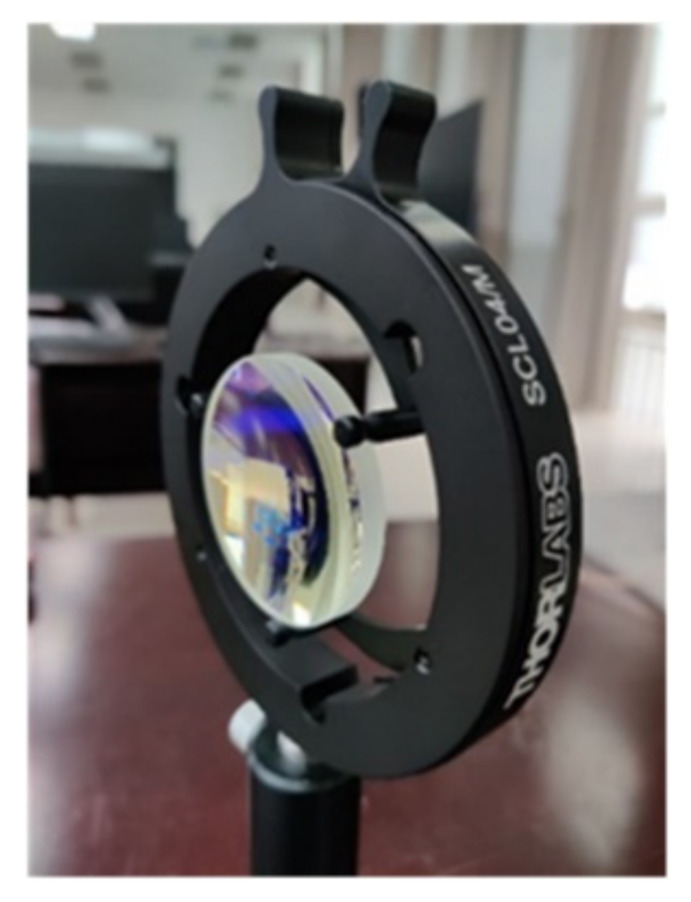
AL5040-B aspheric lens.

**Figure 11 sensors-22-03725-f011:**
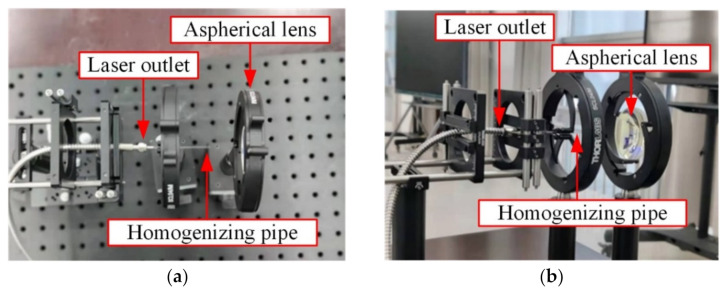
Experimental setup of the illumination system: (**a**) Top view; (**b**) Side view.

**Figure 12 sensors-22-03725-f012:**
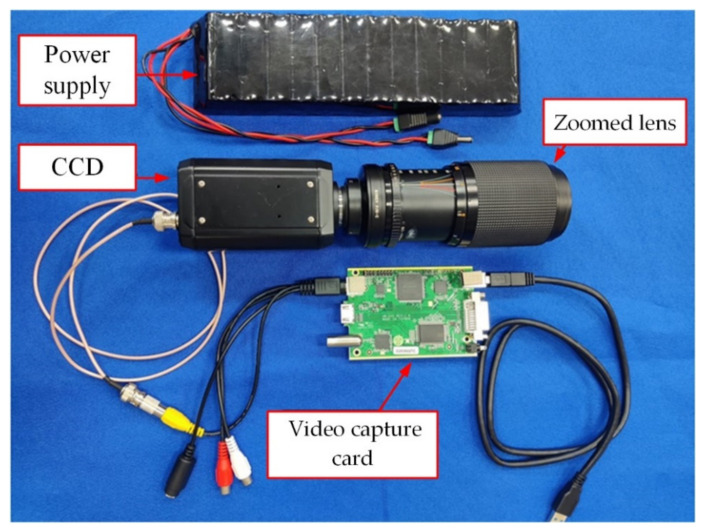
Imaging detection system.

**Figure 13 sensors-22-03725-f013:**
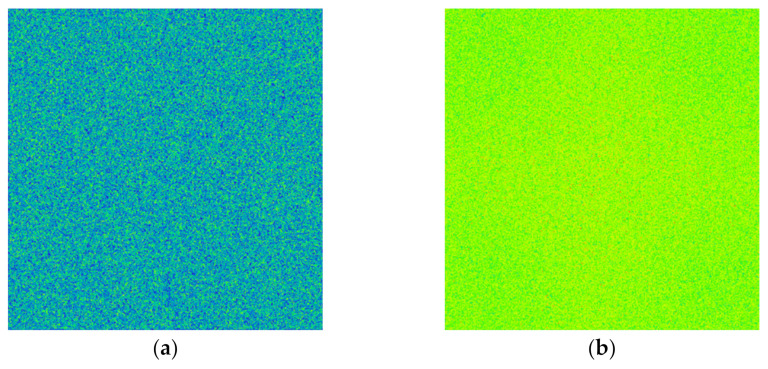
Light spots of different SSRs: (**a**) SSR = 0.25; (**b**) SSR = 1; (**c**) SSR = 3; (**d**) SSR = 5.

**Figure 14 sensors-22-03725-f014:**
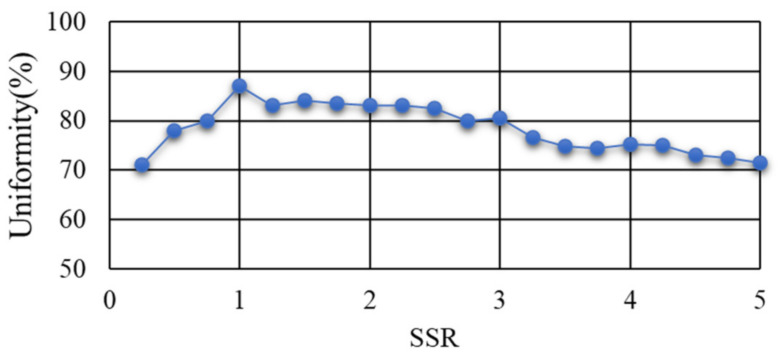
Variation curve of uniformity with the SSR.

**Figure 15 sensors-22-03725-f015:**
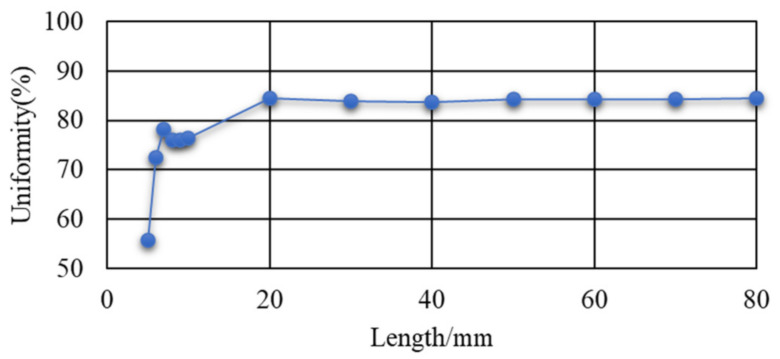
Variation curve of uniformity with the length.

**Figure 16 sensors-22-03725-f016:**
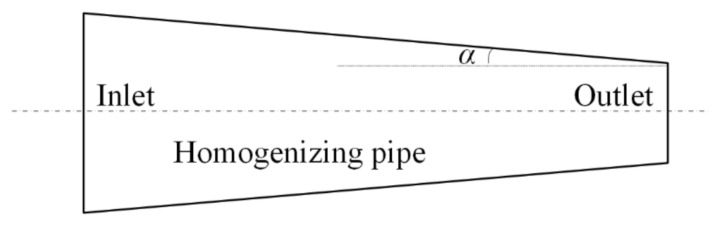
Special-shaped homogenizing pipe.

**Figure 17 sensors-22-03725-f017:**
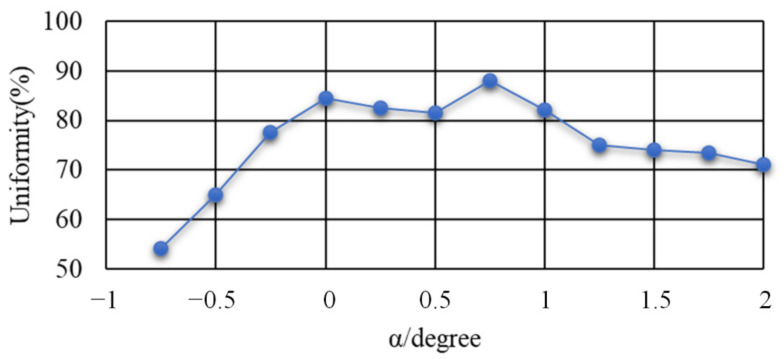
Variation curve of uniformity with the taper.

**Figure 18 sensors-22-03725-f018:**
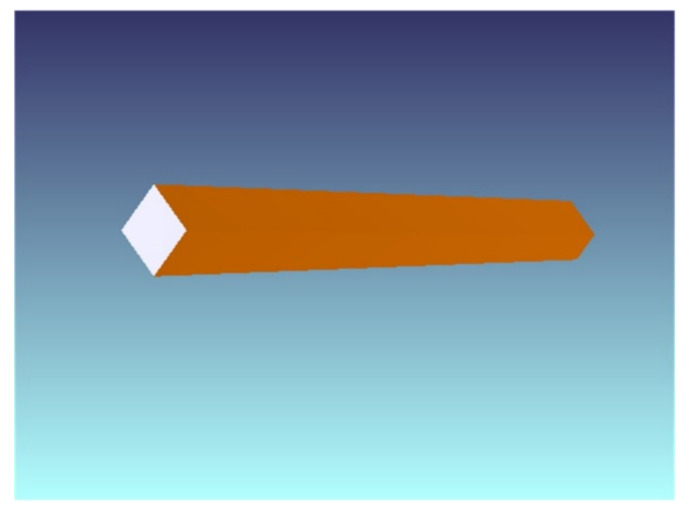
Shaded model layout of the homogenizing pipe.

**Figure 19 sensors-22-03725-f019:**
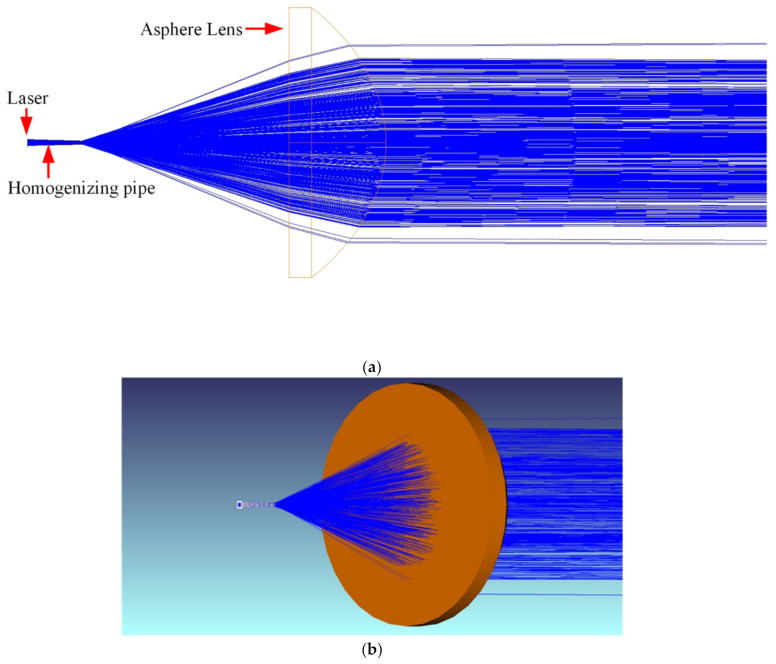
Simulated illumination system: (**a**) 2D layout; (**b**) Shaded model layout.

**Figure 20 sensors-22-03725-f020:**
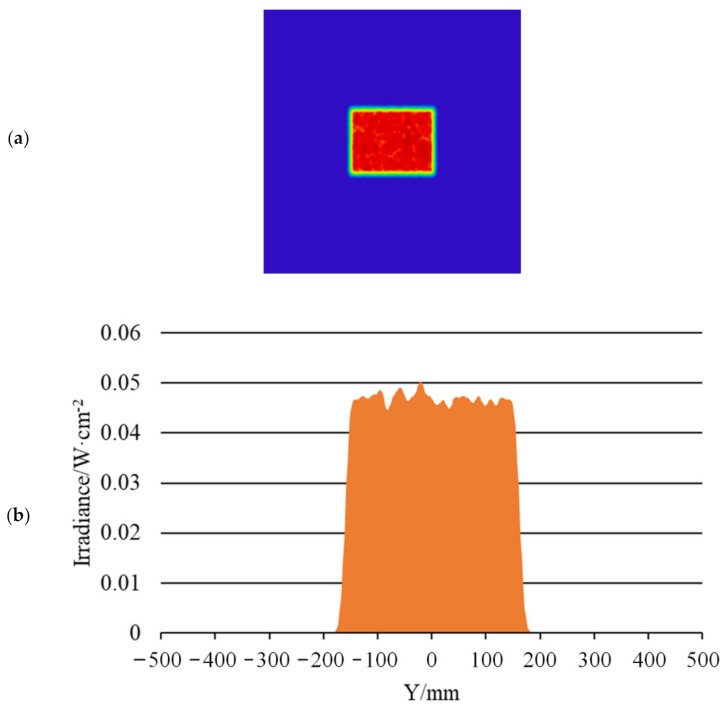
Laser spot and energy distribution at distance of 20 m: (**a**) Light spot; (**b**) Energy distribution in the horizontal direction; (**c**) Energy distribution in the vertical direction; (**d**) 3D energy distribution.

**Figure 21 sensors-22-03725-f021:**
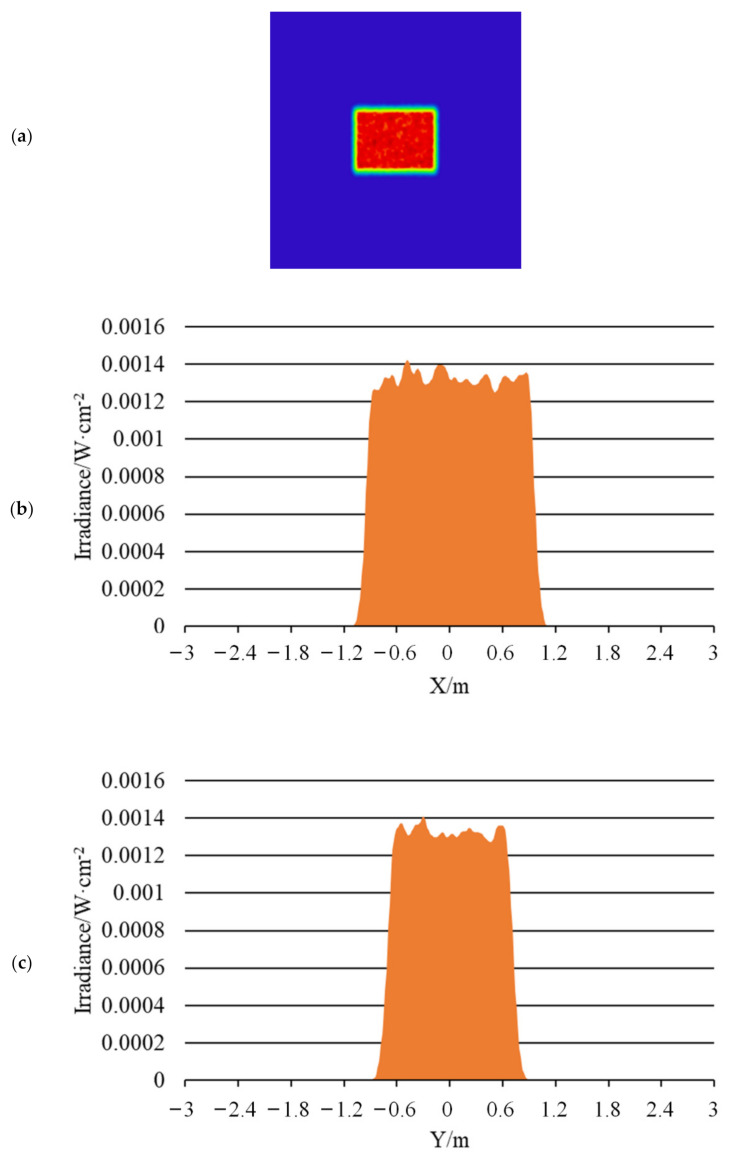
Laser spot and energy distribution at distance of 120 m: (**a**) Light spot; (**b**) Energy distribution in the horizontal direction; (**c**) Energy distribution in the vertical direction; (**d**) 3D energy distribution.

**Figure 22 sensors-22-03725-f022:**
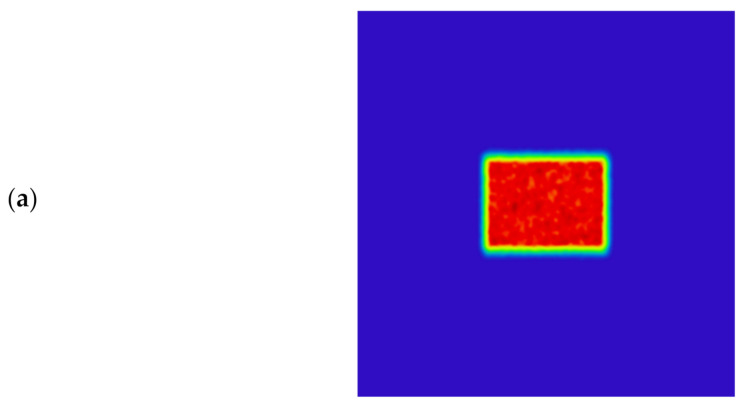
Laser spot and energy distribution at distance of 500 m: (**a**) Light spot; (**b**) Energy distribution in the horizontal direction; (**c**) Energy distribution in the vertical direction; (**d**) 3D energy distribution.

**Figure 23 sensors-22-03725-f023:**
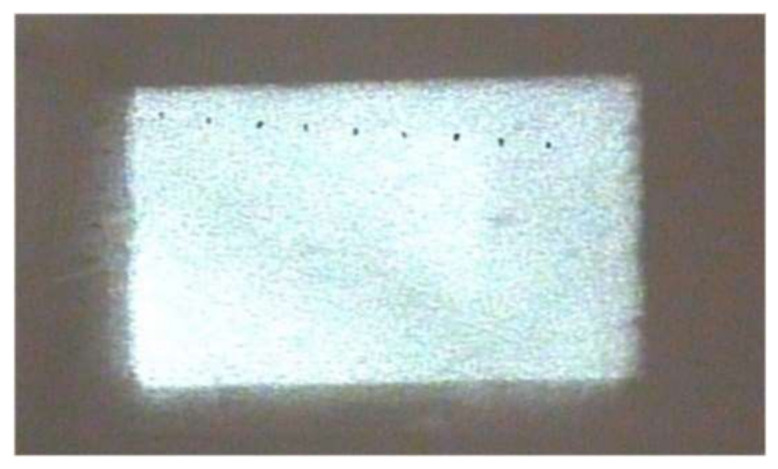
Light spot image at the distance of 20 m.

**Figure 24 sensors-22-03725-f024:**
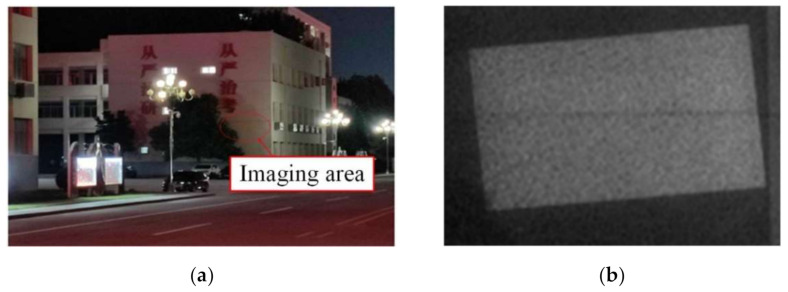
Wall and light spot at a distance of 120 m: (**a**) The Flat large wall; (**b**) Light spot.

**Table 1 sensors-22-03725-t001:** Parameters of the two homogenizing pipes.

Type	Inlet Size	Outlet Size	Length	Material
pipe 1	2 mm × 2 mm	1.6 mm × 1.2 mm	20 mm	Silica
pipe 2	2 mm × 2 mm	1.6 mm × 1.2 mm	50 mm	Silica

**Table 2 sensors-22-03725-t002:** Transmittance of the homogenizing pipe and aspheric lens.

Element	Power of Laser Outlet	Power through Element	Transmittance
pipe 1	0.213 W	0.176 W	82.6%
pipe 2	0.213 W	0.158 W	74.2%
AL5040-B	0.213 W	0.184 W	86.4%

**Table 3 sensors-22-03725-t003:** Experiment results compared with simulation results (20 m).

Results	Uniformity	Diagonal Length	Divergence Angle
Simulation results	85.6%	0.2 m	10 mrad
Experimental results	87.7%	0.26 m	13 mrad
Relative error	2.4%	23.1%	23.1%

**Table 4 sensors-22-03725-t004:** Experiment results compared with simulation results (120 m).

Results	Uniformity	Diagonal Length	Divergence Angle
Simulation results	91.5%	1.2 m	10 mrad
Experimental results	93.3%	1.8 m	15 mrad
Relative error	2.4%	33%	30%

## Data Availability

Not applicable.

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
