# Peer review of "Simulation and Experimental Research on a Beam Homogenization System of a Semiconductor Laser"

_sensors, 2022, doi:10.3390/s22103725_

Round 1
Reviewer 1 Report
In the manuscript, the authors present the simulation and experimental studies on the homogenizing effect of homogenizing pipes with different parameters, i.e., aperture size, length and taper. Both simulation and experiments agree with each other. Basically, I recommend its publication after the following comments are well addressed.
- A large amount of experimental results is given in the manuscript. However, it lacks a comprehensive comparison with the previous results of homogenizing pipes. I think it is necessary to make a fair comparison.
- Line 52: what does the irreplaceable advantage mean? It should be explained in detail.
- In the introduction part, the authors claim that simplicity is the advantage of the homogenizing pipe. I wonder if there are any other advantages?
- It would be good to clearly indicate where are inlet and outlet in Figure 16?
- There are too many figures in the manuscript. I strongly suggest the authors remove some less important figures and/or group some figures.
Reviewer 2 Report
This paper has investigated the influence of geometric parameters of the homogenizing pipe, which will contribute to designing a laser imaging system. Overall, I’m satisfied with the quality of this work. Extensive simulation and experiment results have been provided and appropriate interpretation/analysis are given. I do recommend publishing and hope the authors can consider the following two points.
On page 2, lines 58-83, three beam homogenization methods were listed, which method is related to this work? I understand the authors intend to provide background info, but it should be closely related to the work.
Section 2.3. the name of this section is not appropriate. This section is more about discussing the important design parameters.
Reviewer 3 Report
This paper reports the influence of the geometric parameters of the homogenizing pipe on its homogenizing effect.
Such detailed characterization of the homogenizing pipe will be useful for various applications of the homogenized pipe.
1. However, the homogenizing pipe is a well-known device and technique.
The author needs to explain the originality of this research and the newly revealed knowledge clearly.
Under certain conditions and settings, the result becomes new knowledge, however, that knowledge is only applicable to very limited systems.
Therefore, more generalized knowledge needs to be clarified.
In addition, please consider modifying the paper according to the comments below.
2. Figures 4, 6 and 7 are shown to illustrate the schematic functions and characteristics of the device, however, the figures shown are based on the results of numerical simulations. Therefore,
Please show the fibre diameter in Fig. 4.
Please show the width and length of the homogenizing pipe in Fig. 6 and 7.
3. In sections 3.1.3 and 3.1.4, although the author concludes that the sufficient length is 20 mm, it may depend on the input/output aperture size of the homogenizing pipe.
From the viewpoint of a general discussion, please describe that the length depends on the system under consideration, or please show the other results for different pipe specifications.
4. About Figs. 20, 21 and 22, the authors show the beam divergence angle. Please also show the beam size just after the aspherical lens. It corresponds to a transmission distance of 0 m.
5. Line 331, the distance may be 120m.
6. For Figs. 20-22, please discuss the uniformity values dependence on transmission distance. In fact, from the results of 20m, 120m and 500m, it seems that the uniformity is improved by increasing the distance. Is there an explanation for the mechanism? Please discuss the mechanism.
7. For Table 2, please explain the loss mechanism.
Round 2
Reviewer 1 Report
The authors have addressed my concerns and I recommend its publication in Sensors.
Reviewer 3 Report
This paper reports about the design and experimental evaluation of beam homogenization systems based on the homogenizing pipe.
The basic mechnism of the homogenizing pipe has been well-known and there are some practical applications.
In this paper, some specific applications of the homogenizing pipe are reported in detail and design guideline were discussed.
Such information on the specific applications and specific conditions are sometimes useful for practical applications.
From the viewpoint of such specific applications, this paper contains meaningful information for publication as a journal paper.